# The Valorization of Italian "Borghi" as a Tool for the Tourism Development of Rural Areas

**Carmen Bizzarri** [1,*] **and Roberto Micera** [2]

1    Department of Human Science, European University of Rome, 00163 Rome, Italy
2    National Research Council (CNR) Institute for Studies on the Mediterranean (ISMed), 80134 Naples, Italy;
     roberto.micera@ismed.cnr.it
*    Correspondence: carmen.bizzarri@gmail.com

**Abstract:** The paper comes from the need to search for criteria useful for the valorization of heritage towns, located in rural and/or inland areas of Italy, now affected by depression and depopulation process. To this end, the authors point out how territorial identity can constitute the theoretical foundation to influence development policies and, in particular, tourism development for the sustainability process. It was therefore decided to interview a number of stakeholders who could contribute, with their professionalism and expertise, to identifying possible paths and processes for the enhancement of these areas for tourism development. The methodology was based on in-depth interviews, which allowed for the identification of a of a Strength, Weakness, Opportunities and Threat (SWOT) analysis, offering a guideline for the correct governance of these rural areas for their tourist enhancement in terms of the sustainability of development and tourist attractiveness. The study is an observatory that will monitor the implementation of sustainable tourism enhancement of the "borghi".

**Keywords:** "borgo"; tourism development; rural area; sustainable tourism

## 1. Introduction

The paradigm of sustainability, as known in the literature, can be declined in different areas, being directed to hand over to future generations the resources that today we may have available in quantity and quality. The attention, protection, and care of resources that is evoked in the international arena come from the exploitation that occurred and was perpetuated in the nineties of the last century and has affected all territories and especially those where there has been massive urbanization and population density. Other territories, particularly those located in some rural and inland areas, have been neglected by housing policies and consequently abandoned as they do not possess the typical characteristics of large cities, such as being close to the coast, accessibility, having much more water resources, and being the center of trade. Just in these areas today, the state of abandonment has made their conditions even more worrying both for the purposes of the cohesive and sustainable development of the country, and for the recorded degradation in terms of natural resources but also human resources.

The state of backwardness of these areas today has emerged thanks to the spread of the pandemic that has pushed many people to move away from urban centers to find in these places open spaces where the contagiousness can be reduced, where they can spend long periods of time, and where they can enjoy an authentic and slow lifestyle away from the hectic life of cities and metropolises [1].

The demand to stay in these places has brought out their unexpressed potentialities, but also their limits, especially considering all the houses that are now abandoned and in a serious degradation state, but if properly renovated, are able to welcome not only tourists, but also new generations. Certainly, in order to make these places hospitable it is not only

necessary to renovate just the houses, but the state of abandonment also concerns a whole series of services and common goods, which are crucial for stable housing.

The backwardness of these areas, in fact, has very deep roots, derived on the one hand both from the abandonment of these places especially by young people, who, having no possibility of high rank training and employment, have preferred to migrate, and on the other hand, by the progressive aging of the population that remained there.

This heavy depopulation has caused a loss of skilled labour and withdrawal from industries and businesses, causing both economic and socioenvironmental degradation and at least the decay of biodiversity and cultural heritage.

As the geographic literature shows, the possible remedies historically followed at least three approaches. The first is the so-called "conservative" approach that suggested keeping a minimum level of services for the population to discourage further abandonment. With the second approach, the "compensatory" one, the definitive departure of traditional residents was accepted but measures were put forward to attract new ones. The third approach, the "multifunction" one, was derived by the overlapping of the concepts of inner area and rurality, both expressing territorial marginality [2]. However, all three approaches have failed to motivate the remaining residents to improve themselves—agricultural multifunctionality has not made these areas competitive and innovative—and as a result, these areas are in an even more pronounced state of decline today than they were in the 1990s.

If this process of decay began since the second half of the last century, in the Middle Ages or the Renaissance up to the nineteenth century, they were vital centers, pulsating with economic activity and trade.

Restoring these territories back into dynamic and attractive towns is certainly a long-term process due to the current conditions, which on the one hand may be attractive to tourism due to the conservation and the authenticity—consisting of cultural and social identities, traditions, memories, intangible connections, local peculiarities, and rural landscapes—of their tangible and intangible cultural heritage, but on the other hand are still a long way from sustainable and technologically advanced development capable of attracting investment [3].

Most of these towns have been identified in Italy and are so-called "borghi", characterized by a maximum of 5000 inhabitants and—as written in Ministry of Cultural Heritage and Cultural Activities and Tourism (MIBACT) directive n. 555/216—"by a precious cultural heritage, whose preservation and enhancement are factors of great importance for the country system as they represent authenticity, uniqueness and beauty as distinctive elements of the Italian tourist offer".

Today, they constitute the backbone of Italy, covering a territorial surface of about 54% of its total territorial surface and in some regions reaching even 70–80% of the total regional surface. This expanse of surface does not correspond to the anthropic aspect since the resident population is 17% of the total Italian population, and in some regions, this average percentage still drops compared to the regional resident population.

These towns have wide margins of residential accommodation in a landscape strongly marked by agricultural production, from vineyards to olive trees, from farmhouses in the plains to mountain pastures, and from pastures to terraces, which counteract the degradation and hydrogeological instability.

Many of these towns are characterized by a rural landscape and many others are located in the mountains or on the coasts. However, all of these are very different in their resources and Heritage, which are rich in content, interconnected and integrated, and together with their strong anthropological characteristics, refer to the culture and lifestyles of the settled communities [3]. For this reason, this paper will investigate if these towns, now so depleted to this economic and social development—so they seem to be villages—can come back as an original town, full of life and able to attract tourism. The element that can give a push to activate a new development and transforming the actual villages into

original and authentic towns is their sedimented territorial identities, an element strictly necessary to tourist development.

The territorial identity, in fact, can be defined "by the combination on a given space of a set of situations, extensions, objects, occurrences/presences", consequently of the community, which has the ability to preserve "the interpretative memory of territorial acting, in project, at both local and global scales, passing through all the intermediate levels" [3,4].

The complexity of the process of territorialization is not adequately considered when addressing the policies of the enhancement of the "borghi" for purposes including tourism, but, in fact, thanks to this very process, it is possible to transform that memory and knowledge into a project, since this memorial competence does not concern only the constitutive territoriality but all territorial articulations [4]. The process of territorialization is not only abstract, but provides for the denomination, reification, and structuring that give shape to that project that the community, with its memorial knowledge, identifies.

The paper intends to start with the concept of territorial identity as a tool to represent the different values expressed by the "borghi", such as authenticity, beauty, and uniqueness, and illustrate the fundamental role of residential communities for a new model of tourist development above all in rural areas despite the depopulation and the lack of innovation, increasing the development gap with urban areas [2].

After the pandemic, the slow and open-air lifestyle has launched a process of revalorization of the "borghi" that attracted many smart workers—the people who work online in remote jobs—to become temporary residents and thus changing these communities. This new input into these communities has revealed weaknesses but also opportunities for new tourist destinations.

To deepen this phenomenon, the authors have interviewed some stakeholders, trying to verify what the current sentiments are about the "borghi" and the possibility of going beyond the conservative model, which until now has prevailed in our country, to enhance the "borghi" in terms of attracting new tourist flows. The interest aroused by this mode of study and analysis has given rise to a monitoring unit, which, therefore, will not end with the writing of the paper, but will continue precisely because of the interpretative and trans-sectoral and transdisciplinary fallout that involves the formulation of this new model of tourism in the "borghi".

The paper is structured into five sections. After the introduction, there is the theoretical background section dedicated to the analysis of the main provided contributions in the management and economic geography disciplines on the theme of the "borghi". The third section describes the research method, developed based on the in-depth interviews. The fourth section shows the results based on an integrated reading of the desk and field information collected, according to the "lens" of the consulted scientific literature. Finally, the conclusion section illustrates the theoretical and practical implications, as well as the main limitations and possible future developments.

## 2. Theoretical Background

The enhancement of rural areas cannot ignore a new form of territorialization in which the "borghi" are the main axis from which to start to give coherence and cohesion to the whole Italian system [5].

Recently, the Encyclopedia of Tourism Management and Marketing has included among its entries of "borgo" as a place located in rural or peripheral areas, characterized by authenticity, uniqueness, and beauty, but also social values, traditions, culture and landscape, and the ties and emotional connections between inhabitants and territory—factors that define their territorial identity. The "borgo" is a place where the tourist seeks a recreational, entertainment, and cultural experience, but above all on that is immersive and linked to the encounter with a local reality and a community from which one comes out enriched [6].

The gap that exists today between developed and backward areas, in fact, can be traced, in addition to the geomorphological and infrastructural problems, precisely in the need to reform the community in the "borghi", which, on the one hand, has the memory [3], and on the other hand, is able to interpret the process in place using the most appropriate tools to finalize the efforts towards a participatory and shared production of resources.

In this direction, the "borghi" should be considered "from the perspective of those who live it, experience it, practice it, as well as those who talk about that cannot be disregarded" [7] and ultimately, we cannot disregard the formation of identity, conceived in its processual and dynamic character, with the polysemic implications derived from the different open, complex, and transcalar settings [7] that reflect the peculiarities of the territory itself.

The enhancement of the "borghi", therefore, cannot disregard the formation of the identity itself, as has been expressed in the existing literature—in particular the geographical—directed to outline the determinants of territorial identity, or those environmental factors, those tangible and intangible cultural assets, those socioeconomic trends for which the "borgo" has the ability to produce value thanks to the "localized set of common benefits that produce collective advantages" [8] and the competitive capacity [7].

"It must be emphasized, in fact, that the territory, as well as a universe of experiences, feelings, perceptions, as well as conflicting relationships and geometries of power, is also a concrete entity to be organized and managed, and to make a study of territorial identity useful at the moment of decision-making and planning" [6]. Representing territorial identity means identifying the social system that includes people, traditions, culture, and landscape, keeping in mind the emotional links and connections between the inhabitants and the territory [9].

Ultimately, territorial identity is constantly evolving due to the external and internal agents of the territory that modify its codification as well as the behavioral traits of collective and private action [10,11].

Territorial identity is relationship, a "social construct" [11] consistent with cultural, political, ideological, and ethical processes that is in continuous synergy with cultural heritage, as well as territorial capital [12].

The relationship between cultural heritage and territorial identity [12], in fact, is very close, as cultural heritage and all tangible and intangible assets positively influence local creativity thanks precisely to its very existence and aesthetic values, as well as the visual qualities of the heritage itself, lending itself to many interpretations and many meanings and variations according to the points of view adopted [6,12–17].

This same link should find in the "borghi" the ideal place to express the close relationship that unites the identity with the territorial capital, especially if the territorial capital is defined as a "localized set of common goods that produce collective benefits" [5]. Identity and territorial capital constitute, in fact, that set useful to the development of the competitive capacity of a territory [6] as they clarify both the irreproducibility of each place and their continuous transformation, especially in the "borghi".

The stronger the connection is between identity and territorial capital, the more it will be possible to generate that process useful to enhance resources for tourism activities, and, not only that, but also for the growth of cultural heritage considered as a common good, creating relationality, a common sense of belonging, and the inclination to innovations, creativity, knowledge, coproduction, and cooperation for the constant and dynamic improvement of the local reality, influenced by local empowerment [12] and by the different global interferences [18]. This dynamism can also be expressed by the "presence of multiple territorial identities that render places more fluid, future-oriented, open-minded and able to change, positively influencing economic dynamics" [8], but still having a strong co-science and emotional ties between the inhabitants and their space.

In a study by [8], they identified similarity and solidarity as the determinants of identity: similarity defined as the similarity of physical/geographical status and living conditions, and solidarity as the coincidence and fusion of private interests with collective

ones, originating reciprocity and collective support. Again in the research, which in turn takes up a study developed by [8], it emerges how, in Italy, this identity spirit at the regional level is not very widespread, considering that only seven regions (Abruzzo, Basilicata, Puglia, Molise, Trentino Alto Adige, Tuscany, and Umbria) out of 20 have as their definition an inclusive cosmopolitanism. That is, that model of territorial identity thanks to which the inhabitants, in addition to physical resemblance, have an active and dynamic solidarity so as to be an open community without depriving themselves of their roots and belonging to local resources.

The same research shows that five regions (Calabria, Lazio, Liguria, Marche, and Sicily) out of 20 are characterized by individualistic localism, in which each regional community is united by physical, geographic, and institutional similarities, but solidarity is not practiced.

As has already been written, it is important to point out that territorial identity is dynamic and therefore moves from individual awareness and the collective sharing of experiences of place to channel them into activities and projects on the territory always in progress, avoiding an instrumental use.

The recovery of the "borghi" for tourism purposes, therefore, can be undertaken only if it forms a territorial identity capable of reviving an active and dynamic community, both to activate that sense of uniqueness of places, the genius loci, and the narrative of cultural heritage.

It should be noted, however, that to date, despite the Italian government's so-called "recovery fund" plan and all the current measures useful to the development and economic growth of rural areas and the "borghi", there are not enough economic and financial resources for the recovery of all the 5000 Italian "borghi".

It appears, therefore, a priority to adopt a multilevel governance and a multiscalar approach, in which territorial indicators play a key role in maintaining comparable information to detect the territorial diversity [19]. In this context, the "organizing principle" becomes "geo-graphically" relevant thanks to which we determine the positional, cultural, and functional relationships between biotic and abiotic elements in a "technical rationality" such as to be interpreted with a horizontal process. That is, the participatory and shared production of resources by all stakeholders using the most appropriate tools to finalize the efforts.

This organizing principle can be precisely that territorial identity in which, in addition to environmental, social, geographical, economic, infrastructural, and cultural factors, the values coming from the experience of each resident are included, an experience of which the cultural legacy and the Cultural Heritage are a part. Both of these latter factors (for a definition, see [10]) are a means to set up a reflective citizenship and a high sense of belonging that is achievable when an educational and training process is developed addressed to the younger generations, who will thus feel they are active participants in their own identity in sensory and cultural terms.

Building this consciousness in young people is essential to live in the territory as a space, in which each citizen becomes not a simple occupant of the space, but an occupant for that space, determined by the conditions offered not only in terms of geography and environment, but also by the historical, social, and cultural load that has configured it as such.

If each subject gives its own meaning to the space, it will be possible to transmit diversified narratives, during which the subjective elements are mixed with the objective ones. This narrative becomes, therefore, identity when the images and morphological, spatial, functional, aesthetic, and cultural traces are represented, no longer subjective, but objective, and thanks to which each component of the local community identifies itself.

Although territorial identity is mainly characterized by social factors (see the previous paragraph) as can be seen from the developed scheme, it is achieved when all the environmental, economic, and infrastructural elements become an experience belonging to the community. The theory of community-based tourism [20], in fact, is based on the sharing and the participation in the planning and programming choices of the tourism

product: the local community, thanks to spatial proximity, trust, and mutual interest, can easily create services and networks using not only natural, environmental, and cultural resources, but also human resources and the most appropriate technologies of the area. Solidarity and geographical, economic–social similarity are, in fact, the synthesis of the processes of territorialization, spatialization, and reification of the community towards space and society itself [3]. The tourist valorization can be effective only if all communities participate in this process for the finding of authenticity and the differentiation of tourist supply: Each local community has its history and its material and immaterial cultural heritage narrated with storytelling and becoming the tourist experience during the visit of the "borgo" [20]. The participation in the process of valorization from the local community to all private and public stakeholders in the coproduction of tourist products can affect the tourist experience and reduce the probability of exceeding the carrying capacity and the effect of demonstration: The residents pay attention to the external diseconomies (rumors, waste, pollution, and congestion) and create a sustainable tourism system, also as the tourist supply is direct to niche tourism [21]. This segmentation of demand is naturally formed, and for the hospitality system so that in the downtown of the borgo is not too countless and so the tourist flow is very interested to this territorial supply, avoids mass tourism. Therefore, in a lot of these "borghi" the tourist can visit and participate in many activities as such as: visiting equestrian centers and horse riding, winemaking, revived historical festivals, handmade pottery, and so on. However, all these activities are only tourist experiences when the residents tell the story and help the tourists to test these activities.

For this reason, the engagement of residents, above all in this moment when the territorial identity is so fragile from the depopulation and depletion, are very important to planning the tourist valorization through the coproduction, producing together tourism services from their conception to their implementation [22]. These sharing activities will be realized from all public and private stakeholders that develop a tourist system based on a new balance of local resources and by pursuing the following fundamental objectives for the sustainability of development:

-   A "smart" growth policy, directed towards efficiency and appropriateness in the use of resources thanks to investments in the high-tech sector and innovation. A "borgo" can be a smart destination as a "an innovative space, accessible to everyone, and consolidated on a cutting-edge technological infrastructure that guarantees the sustainable development of territories, facilitates the integration and interaction of visitors with the environment, and increases the quality of their experiences at the destination as well as residents 'quality of life" [23] without destroying or changing the territorial identity or the cultural heritage, but finding a balance between all of these elements.
-   An "inclusive" policy, characterized by the enhancement of local human capital, including those coming from peripheral internal and external areas, those who are economically disadvantaged, or those with special needs for integration.
-   A policy aimed at the development of the "green economy" aimed at an appropriate use of resources through a proactive capacity of regions and cities that develop strategies to support the prevention and protection of the environment.

The development of the "borghi" for tourism purposes, therefore, requires the activation of a "continuous process of co-evolution between human society and environmental re-sources" [3], that "system of rules and attitudes related to local culture and history, aimed at achieving individual and collective goals" [18], and that social capital that enhances the economic and social integration. It is about acting on those "intangible elements, whose endowment presents a precondition for the valorization, management and transformation of external shocks into opportunities for internal development" [24].

Surely the latter alone cannot, as already noted [16,25], trigger those mechanisms of development, but requires the convergence of external policies, founded moreover on the centrality of the person [26] and the needs of the community, realizing inclusion as a model

of innovation "through a meaningful participation of the community and respect for the values it expresses" [14,27].

Finally, social innovation can be one of the main components of the transformation of the "borghi" into smart destinations for a sustainable tourism, thanks to technologically advanced solutions such as to respond to a tourism which, as defined by the World Tourism Organization, is "clean, green, ethical and quality at all levels of the service chain" [28–30], thus minimizing the use of natural resources.

## 3. Research Method

The path followed in the research has been descriptive, since its aim is the identification of certain results, following the study of reality [31–33].

The empirical research was divided into a desk phase in which documents and information were collected from websites and related to cases of "borghi" that have already undertaken paths of tourism development that have benefited local economies and a field phase in which in-depth interviews were administered to 11 key informants (*KI*) chosen from among academics, professionals, and experts on the evolutionary trajectories of the "borghi".

The choice of this method is linked to the exploratory nature of the research objective. The intention was not to generalize the results, but to gain in-depth knowledge of a new phenomenon by accessing the perspectives of the interviewees, capturing their mental categories, interpretations, opinions, perceptions, and their reasons for their actions [27].

In the first phase, data were collected by combining documents and semi-structured interviews. Documents used for systematic evaluation as part of the study took a variety of forms, including background documents, brochures, journals, event programs, letters and memoranda, press communications, institutional reports, and information available on the websites of municipalities where revitalization projects were developed.

Among these documents, the main were:

- The "Charter of the Borghi", in which the Italian Ministry of Tourism has included the "borghi" as a tool to renew the commitment to a sustainable development of the territories and is aimed primarily at the enhancement of the internal areas of the country.
- The "Save borghi" law, for the support and enhancement of small towns up to 5000 inhabitants. This law, starting from the need for an alliance between innovation, territorial cohesion, and beauty as the basis of the future of Italy, proposes a series of qualifying measures to lay the foundations of an economy more on a human scale that focuses on communities and territories, on the interweaving of tradition and innovation, old and new knowledge.
- The action of the "National Plan for Recovery and Resilience", called "Tourism and Culture", which contains a series of measures for the protection and enhancement of architecture and rural landscape and for the interventions of restoration and rede-velopment of rural and historic buildings. In this sense, the Plan aims at a revival of small towns, largely abandoned, and saves the immense rural building heritage composed in Italy of huts, farms, farms, and stables at hydrogeological risk, creating the conditions for the revival of these small municipalities.

To these documents must be added the information provided on some generalist websites such as "I Borghi più belli d'Italia" (https://borghipiubelliditalia.it/ (accessed on 18 May 2021)), the initiative created thanks to the contribution of the "Tourism Council of the National Association of Italian Municipalities (ANCI)" with the aim of enhancing the great heritage of history, art, culture, environment, and traditions present in small Italian towns that are, for the most part, marginalized by the flow of visitors and tourists and "Il borghista" (https://www.ilborghista.it/ (accessed on 20 May 2021)), the portal dedicated to villages and tourists in villages.

Finally, there are the municipalities' strategic planning charts that have implemented a set of policies and actions to enhance their "borghi", contributing to the development of

tourism in the area. In particular, there are the strategic development plans of Civita di Bagnoregio (https://www.comune.bagnoregio.vt.it/ (accessed on 20 May 2021)), that of Santo Stefano di Sessanio (http://www.comunesantostefanodisessanio.aq.it/hh/index.php (accessed on 20 May 2021)), Spello (https://www.comune.spello.pg.it/ (accessed on 20 May 2021)), Recanati (http://www.comune.recanati.mc.it/ (accessed on 20 May 2021)), Santa-Fiora (http://www.comune.santafiora.gr.it/hh/index.php (accessed on 20 May 2021)), and Buriano in the Municipality of Castiglione della Pescaia (https://comune.castiglionedellapescaia.gr.it/ (accessed on 20 May 2021)).

Moreover, this information has been useful to design the interview guide, and researchers have analyzed the documents in order to obtain basic information and useful elements for the definition of the questions to be asked during the field research.

The second phase, carried out from February 2021 to March 2021, consisted of semi-structured interviews with 11 key informants:

(1) The President of Cultura Italiae (Key Informant 1—KI1);
(2) The Mayor of Lanusei (Key Informant 2—KI2);
(3) A Village Innovation Manager (Key Informant 3—KI3);
(4) An Expert on Digital Tourism Applications (Key Informant 4—KI4);
(5) A University professor of economic geography with expertise in "borgo" development (Key Informant 5—KI5);
(6) A journalist and author of essays on "borgo" regeneration (Key Informant 6—KI6);
(7) The magazine director of "Italy's Most Beautiful Villages" (Key Informant 7—KI7);
(8) An agronomist expert on the recovery of internal areas (Key Informant 8—KI8);
(9) A former Assessor of Tourism for the City of Syracuse, expert in territorial marketing (Key Informant 9—KI9);
(10) An expert in tourism and cultural heritage enhancement (Key Informant 10—KI10);
(11) A financial analyst, expert in the regeneration of "borghi" (Key Informant 11—KI11).

The interviewees were identified thanks to the contribution of the Culturae Italiae team of, a platform at the service of the country to offer a concrete support to the definition of a common and collective space of shared civic and social design commitment, whose aim is to develop and disseminate a cultural model, sustainable and competitive, able to promote a cultural change in the community and in public administration, governed by ethics and design intelligence.

From this platform has been generated a spin-off committed to the theme "Recupero borghi italiani", in which academics, researchers, professionals, and experts involved in various capacities on the subject of the villages participate. These teams periodically discus the social, cultural, practical, regulatory, economic, and visionary aspects of the recovery of Italian "borghi".

A key feature of semi-structured interviews is that they provide in-depth information about a certain phenomenon covering various issues concerning the study. The inform-ants were selected according to a theoretical sampling criterion [28], looking for the adequacy of the sample rather than its representativeness.

The interview aimed to collect opinions from different key informants in the recovery of Italian "borghi" and to identify shared criteria useful for the rebirth and enhancement of these little towns for the recovery of post-pandemic tourism in our country. Starting from this scope, as always happens in the application of this research method, it was also possible to verify the accuracy of the information in our possession and to obtain new information, to make the research project known and involve the interlocutor, and finally, to better define and build our project proposal [34].

In this regard, in the sample we have included participants who had an in-depth knowledge of the phenomenon of the villages, of the internal areas, and of the opportunities associated with the development of tourism.

The objective of the interview was to get the point of view of key informants on the role of the "borghi" in the Italian context for a new tourist offer and as an engine for the valorization of the internal areas of the country.

The preparation phase involved defining the interview outline consisting of 11 open-ended questions. However, it should be noted that the interview was a conversation provoked by the interviewer addressed to subjects chosen based on a survey plan with cognitive purposes guided by the interviewer on the basis of a flexible and non-standardized interview guide.

The interview outline included the following topics: the concept of villages; their role in the recovery of tourism; the possible tourism vocations of the "borghi" and the people interested in using them; the new tourism targets; the possible strategies and actions to be undertaken to revitalize the "borghi"; the tourism services to be implemented; the possibility of using recovery funds; accessibility and the use of new technologies; the threats, opportunities, strengths, and weaknesses of the Italian "borghi"; and finally, the contribution of the "borghi" to the development of the internal areas of the country.

In particular, the following is the scheme of questions:

1. Could you provide us your definition of "borgo", emphasizing its main distinctive factors?
2. In your opinion, what could be the role of the "borgo" for the recovery of tourism?
3. What could be the tourist vocations and the people interested in using them?
4. What flows and typologies could a village or a network of connected "borghi" intercept?
5. What do you think are the strategies and actions to be put in place to revitalize the "borghi"?
6. What are the basic tourist services and systems that a "borgo" should offer?
7. The recovery fund has allocated 1.5 billion euros for rural areas; how can these funds best be used to enhance the value of "borghi"?
8. Do you have any ideas for accessibility-also to realize "tourism for all"-and internal mobility?
9. What is the role of application of new technologies (energy efficiency, telework, telemedicine, mobility, building renovation, home automation) for the recovery and development of "borghi"?
10. What are the strengths and weaknesses of Italian "borghi"? What are threats and opportunities for "borghi"?
11. The "Borghi" are often located in inland areas of the country, so how can they contribute to the systemic development of the entire area?

In addition, as part of the preparation, prior to submitting the interview, we documented the role and initiatives of the stakeholders.

Finally, an agenda of appointments was defined for the realization of the interview through the Google Meet platform.

In order to pursue data saturation: (1) the interviews were structured to facilitate asking the same questions of the participants; (2) we constructed a saturation grid, in which the main issues and topics related to the conceptual framework were listed on the vertical axis and the interviews were listed on the horizontal axis [35]; and (3) we relied on triangulation of data across multiple sources [36].

Finally, considering the authorization received from the interviewees, the interviews were recorded and then transcribed word for word, keeping the entire content unchanged.

## 4. Results and Discussion

The first part of the survey is focused on the concept of the "borghi" in order to understand whether the definition used by MIBACT could be considered exhaustive in the operation.

The Minister of Heritage and Culture and the Tourism Directive of 2 December 2016 n. 555 "2017—year of the Italian borghi", in fact, considers the borghi "the Italian municipalities with a maximum of 5000 inhabitants characterized by a valuable cultural heritage, whose preservation and enhancement are factors of great importance for the

Country System as they represent authenticity, uniqueness and beauty as distinctive elements of the Italian offer".

A large part of the interviewees underlined how this definition, despite placing emphasis on distinctive factors of the "borghi", such as valuable cultural heritage, authenticity, uniqueness, and beauty, should have emphasized the need for the human and social components, considered the basis of a community representative of the identity of these territories.

The territorial identity becomes substantial in the definition of the "borghi", discriminating history and the distinctive feature of the "borgo" itself, placing it between past, present, and future, as a place of life and growth, activity and work.

The elements of landscape, already recalled in the European Landscape Convention, become characteristic of these "borghi". Types of natural green elements and not and the methods and agronomic techniques consistent with the geographical and climatic characteristics are those that give the identity of that place.

In a prospective view, the dimensions to be included are tradition, customs, and traditions, as well as all the elements pertaining to the cultural identity of a historicized place, even if they are recent but historicized enough to make them "precious".

The historical center becomes one with the surrounding heritage. This is the key to understanding, and the community is a fundamental part of it as it unites the historical territory to the innovative one, which is also expressed by new technologies and skills, as well as by the real needs of the "borgo" itself.

A place where tourism can be developed always with a view to safeguard the territory, understood in its human, historical, architectural, natural, landscape, cultural, social, and economic components.

> *"The theme of 'beauty' must be accompanied by historical, cultural and landscape identity."* (KI2)

In addition, in order to highlight the touristic value of the "borghi", one should also consider their touristic vocation, measured by the presence of touristic services and a tourism offer system in the area.

Although in the definition there is the word "uniqueness", it is not so clear what it actually means, since the "borghi" are heterogeneous depending mainly on their geographical location. This geographical aspect is not explained at all in the definition, while it represents a distinctive feature and is its origin.

From the interviews, it emerges that "borghi" can be anything but unique. There are "borghi" with a tourist value and "borghi" with other vocations. The municipalities that do not have tourist attraction resources should devote themselves to other vocations, for example industrial, agricultural, and activities not necessarily dedicated to the tourist aspect. In this way, special vocations should be enhanced.

> *"The suburb is a cultural work, which lives of pluralisms."* (KI11)

> *"They are very heterogeneous entities, because there is a difference between 'borgo' and hamlet, because there is a difference between the neighboring hamlet and the hamlet of inland areas."* (KI5)

The main limits of the definition provided by the MIBACT are found in the identification of a number (5000) of inhabitants that qualifies the village and in the identification of extremely subjective parameters.

> *"The numerical aspect is certainly not a parameter that qualifies, it has only administrative, regulatory and legislative value."* (KI4)

In any case, it must be said that definitions have a temporary value in the sense that they should be historicized, because they photograph a specific phenomenon in a specific historical moment.

> *"Any definition should always be considered in relation to objectification with the identification of indisputable parameters."* (KI1)

With respect to the theme of the role that the "borgo" can assume for tourism revival, the participants consider the "borgo" as a reality that grows from local sentiment, and tourism is only a decisive part of this, but one that was introduced later.

> *"'Borghi' express great potential for relaunching tourism, but tourism cannot be the only economic sector to be relaunched."* (KI3)

In this sense, a "borgo" has a strategic role in economic recovery and is a fundamental part of the process of systematically enhancing the country. Therefore, it is necessary to first make these places livable and improve their housing quality.

These municipalities form the Apennine ridge, and it is necessary to give them a voice since one of the main risks is the maintenance of the territory itself. It is essential to create the conditions that favour residency. The "borghi" must be repopulated in the genetic values that have formed them and that are those of the "culture of doing" linked to craftsmanship, agriculture, food, and wine.

Only then can we think of developing them in terms of tourism. After all, a tourist is nothing more than a temporary citizen and, as such, needs to live a visiting experience in line with the standards of the place he goes to.

> *"On the other hand, tourism comes when the residents live well, not the other way around."* (KI10)

In this historical moment, "borghi" for tourism take on a dual significance. They are, in fact, suitable places for proximity tourism, but also useful in this phase of coexistence with the COVID-19 pandemic.

At the same time, once this contingent situation has been overcome, it will be important to bring attention back to these small but valuable places, which have the possibility of distributing tourism flows more efficiently over time (seasonal adjustment) and space (decongestion).

Following the COVID-19 pandemic, tourism will be totally disrupted, both from the demand side and the supply side.

The "borghi" are destinations and factors of attraction for a more specific target. Therefore, in a scenario of conspicuous investment in infrastructure and greater sensitivity and entrepreneurial will, many "borghi" can be linked to tourism and enter fully into the world of hospitality.

> *"Italy is an open-air museum made up of many small towns of great value and quality. Italy of the 'borghi' is a hidden Italy, different from the one that everyone in the world knows (Venice, Milan, Florence, Rome, Naples, etc.)."* (KI7)

In order for the "borghi" to assume a strategic role in tourism, it is essential to have a systemic vision of the overall dynamics of tourism itself. It is necessary to have an overall strategic approach that flows into a planning of what one wants to achieve over time.

The community that lives in the "borgo" is the engine of this valorization, since it represents the factor of attraction for a tailor-made tourism, not mass tourism, attracted by an innovative offer rooted in the authenticity of the territory and in the quality of life.

In this way, the "borgo" can become the "connection node" of a territorial matrix. The "borgo" is the point of consolidation of an experience that is made by passing through the territory.

> *"Between 'borghi' and 'borghi' there is the journey, the narration, the transfer, the visit experience."* (KI8)

It represents the part of the tourist experience that guarantees the connection between the practices of handcraft, agriculture, environmental components, food, etc. In this context, even the architectural structures are functional to hospitality while also becoming an experience.

It would be desirable, therefore, to transmit to the traveler a predominantly human experience that makes the tourist breathe an atmosphere characterized by social relations, food, and handicrafts and not necessarily enjoy it in the form of prepackaged tours.

In the "borgo", it is necessary to find elements that correspond to the lifestyle of the residents. In this way the "borgo" is alive and the tourist is triggered in a mechanism and in a place that is alive in itself.

From this, we understand how the "borghi" tourist vocations are linked to health, beauty, and well-being, but without neglecting the theme of hospitality.

It is a context that, in terms of visiting experience and methods of use, has strong peculiarities and is profoundly different from that of cities of art.

The "borghi" are not places suitable for hit-and-run tourism, but a slow tourism that allows you to enjoy, in addition to the artifacts, the beauty of the works and the landscape, the relationship with a welcoming community.

That of the "borghi" is a cultural tourism in which the human and relational dimensions play a priority role.

Starting from these, the subjects interested in enjoying them are transversal. The common characteristics are the interest in wanting to have an immersive experience and a close proximity.

They are ageless subjects, curious about history and nature, interested in the discovery of others and who, at the moment of the journey, intend, even temporarily, to reset their previous condition and place themselves in a totally new condition.

*"Sportsmen, historians, food lovers. People ready for any discovery."* (KI8)

In a traditional vision, the "borgo" is a place suitable for a walk-on target, the excursionist, who needs hospitality for a limited period.

However, there is also a definable "residential" target: tourism from an average stay of 6 to 10 days or, at different times of the year, residing intermittently, or even intending to move in permanently.

Both are targets that need to relax, and who benefit from open spaces in contact with nature.

Today, the pandemic period associated with the growth of remote work has generated new trends in demand, the so-called digital nomads. People that improve their productivity and enhance the quality of working life because they immerse themselves in different rhythms in more welcoming environments (see the farmhouse in a rural area). At the same time, it is possible to set up a space for co-working and co-living, therefore working in shared spaces (rationalizing costs).

In summary, the possible tourist targets attracted by the "borghi" can be:

- The classic tourist, that is to say visiting for 1–2 days, a maximum of 3 days. They are also defined as a comparative tourist or the excursionist, who stays for a limited time;
- Tourist of the roots, usually emigrant, who returns and stays every 3–6 months a year and who considers the "borgo" the place for his retirement, finding a quality of life superior to his daily life;
- Ageing people, who have more free time and a greater spending power. Although, for these, the appropriate conditions must be created to ensure inclusion and accessibility to avoid confining these people in places where sociability is limited to residents;
- Digital nomads, individuals who work in any location as long as they can use technology;
- young families with children of medium to high income; and
- Young people who have a very strong power of influence on others (they are the fashion bloggers or the influencers who set the trends) and are the forerunners. At this time they are the only ones traveling without too many safety problems.

Strategies to revitalize the "borghi" must start from two key words: convenience and culture.

The community must become aware that enhancing the value of the "borgo" is an opportunity for everyone, and this requires a profound cultural change.

Those who live in the "borgo" must understand culturally that visitors and tourists bring both economic and social wealth.



*"We need to think of the tourist as someone with whom to share these values."* (KI4)

At the same time, it is essential to consider that the valorization of the "borghi" passes through a sustainable management of the so-called commons.

*"Common goods that must be managed as such and not be left to those who contribute to the creation of parasitic income situations."* (KI11)

Therefore, there is a need to create an organizational and governance model that, in addition to being based on public–private collaboration, considers the community as an essential factor. A community that constitutes the humus of this revitalization that is the result of a combination of tradition and innovation, old and new, old and young.

*"The elderly who represent memory, the young who have the time and energy to develop the content of the elderly."* (KI8)

Much of the interviewees emphasize the importance of a process in which the top-down path meets the bottom-up path.

The regional and municipal governments play a key role in the marketing of the "borghi" and, in addition to directing infrastructure investments, they can act as facilitators for those initiatives coming from "below" and from the community itself.

It is important that strategies from above, focused on facilities and the development of access infrastructures and transport, must be matched by strategies from below, contributing to the construction of a common territorial identity.

The logic to be triggered is that of a virtuous circle. If the municipal administration finds that there are a series of initiatives for the purchase of real estate and an attendance of the place, it can become an active part of the process. It can, in fact, sensitize the local population to understand the importance of this type of development and get them to take part in the initiatives of the "borgo".

For example, in the form of community cooperatives: The healthy cooperative model can take charge of a vision and recovery of its own "borgo". The basis of a hamlet's development lies in collective planning.

*"It is essential to raise awareness towards groups of citizens who in a "passionate" way can dedicate themselves to the recovery of a 'borgo'."* (KI5)

The municipal administration must act as a driving force and, if smart, support the real estate regeneration and cogeneration project with a series of initiatives and service activities while also consulting territorial marketing agencies that bring foreign investors.

*"Making international agreements and then activating residents, who represent the trigger factor in this process."* (KI9)

At the strategic level, the role of the central government should not be underestimated, as it can enable the "borgo" to function both as a "museum site", as a cultural "resource of attraction", and as a site with a cultural value, which, thanks to the dynamism of the activities carried out, can enhance the little-known areas of the country.

The development project must be set up and shared with the resident community. The population must be able and willing to accept the development project.

The process must be carefully constructed, with clear objectives and a long-term time horizon.

The actions to be put in place can be summarized in a few key words:

- accessibility
- digitization
- security
- training
- networking
- communication.

With regard to accessibility, digitization, and safety, these actions presume infrastructural interventions linked to investments to facilitate accessibility and internal mobility in these places, the installation of broadband, and the creation of medical care facilities.

Training is a key dimension of this development process. There is a need for a greater culture of hospitality. The latter must be thought of as an industry in the positive sense of the term. In this sense, the best practices of Matera or Grottole must be studied, in which there are real Academies of experiences.

> *"The handcraft must be made to understand that if he were to tell anecdotes or would create a storytelling around the process of making certain products, this would increase the value of the tourist experience."* (KI4)

The action of networking concerns the system of offerings in which the opportunity is created to network the "borghi". On the contrary, individual realities will never be competitive. The single "borgo" is not enough for the area to become a tourist destination. A synthesis operation must be made, creating paths between the various "borghi", and making it clear that Italy is a country of "borghi".

In this sense, the "Cammini project" (Roads project) or the "Via Francigena" (Walking Route) represent cases of very successful networking. The return on image of these initiatives is very strong. What arises around them can be useful in revitalizing the knowledge of these territories. Further work could be done to network the "borghi" and create specific tours and slow tourism. This applies to both inland and coastal areas.

This would involve a more efficient distribution of resources and activities, but it would also profoundly change the dynamics of marketing, moving from the "borgo" to the "borghi".

> *"We are not thinking only of very fortunate, extremely well-known 'borghi'. The question must focus on lesser-known realities and the effort is very strong if one wants to emerge in order to make the territory adequately known (see the case of Tropea)."* (KI7)

In addition, networking associated with digitization would make it possible to fortify the relational dimension and attract new tourism targets: workers who, in contact with nature and a pleasant environment, can benefit from these elements of diversity.

Communication is also an important action in which to invest. We refer to both territorial marketing aimed primarily at international players interested in investing in the "borghi", increasing the appeal, and actions of tourism marketing directed to specific targets, both traditional and new.

In terms of tourist services, those interviewed emphasized that the system of tourism services that a "borgo" should be equipped with are, on the one hand, those essential to welcoming a visitor, and on the other, those necessary to support an experience that best characterizes that "borgo".

The "borgo", at this moment, has two needs: to strengthen the offer of services and to provide the possibility of carrying out activities without being in less comfortable situations. All of the actions aimed at increasing services so that they are on par with the services available in a large urban center are actions that go in the right direction.

It seems essential to create a combination of qualitatively homogeneous commodities, in which it is also possible to align or characterize the various services.

Tourists are always looking to combine the enjoyment of cultural resources and the rediscovery of natural itineraries, which are therefore the factors of a reality in which they are daily distant, with the availability of services that they can normally use.

From the point of view of "tourism for all", it is necessary to foresee interventions to improve accessibility for the various targets of disabled people, studying solutions case by case.

> *"For those coming from the city, it is necessary to create a sort of continuity of services: it is important to know that the tourist will be looking for services he uses daily and at the same time will benefit from the greater space available."* (KI4)

For the purposes of tourism development, accommodation services are essential. At the same time, however, services aimed at making the potential and the peculiarities of the territory known are also indispensable. In this area, maintenance and management services

of buildings should be provided, with specific reference to those dedicated to residential tourism.

Moreover, the "borghi" are often barycentric with respect to a whole series of events that take place in a wider range. Therefore, guides, experts, and professionals who accompany you in the fruition of the territory are extremely appreciated.

As already pointed out, connectivity services are also fundamental for development, but are equally important is the provision of services useful for creating specific itineraries, respecting time and permanence.

*"Ensure a balance between digital services and socialization and assistance services."* (KI5)

Assuming that the inland areas have orographic problems, instability, an absence of health care networks, education, and recently, the growing need for adequate telecommunication networks, in view of the development of the "borghi", the funds allocated by the recovery fund for the inland areas, in the opinion of our interviewees, could be used in the implementation of projects that can increase the appeal of the "borghi" to people who could contribute to the repopulation of these places.

*"Conditio sine qua non is an approach to planning that takes into account a systemic action between several municipalities that network."* (KI2)

Funding, whether public or private, must be managed with a model of good business already thought of by local governments so as to give sustainability to the whole systemic action.

This presupposes a radical change from a definable perspective of requesting welfare to the entrepreneurial management of the "borgo".

This would be the first step towards the productive revitalization of the "borghi" and the beginning of a process of sustainable development.

*"The only thing that would need to be accomplished is to make those areas fit for 2050."* (KI1)

In line with these objectives, priority is given to interventions related to the hydrogeological structure to improve road networks and make accessibility and mobility possible, as the "borghi" are mainly located in the Apennines of the country.

Alongside these interventions, we should consider the investment in broadband, designed as a necessary condition for travelers.

*"Those who visit a place must be able to connect with it, if this does not happen that area will be penalized."* (KI4)

This aspect should not be underestimated since the digital infrastructure for an inland area is totally different from that of an urban center. Technological equipment designed primarily for urban areas is not easily transferable, in terms of cost and infrastructure, to inland areas. It takes a lot of planning and structuring to make this transfer happen.

Part of the funds could be allocated to public–private partnership projects to strengthen the systems of supply and develop greater attractiveness at the international level.

The opportunity highlighted by some interviewees to direct funding towards the creation of schools is very interesting. Education in these areas could play a strategic role in outlining the future vocation of the territories. This applies to the entire educational pathway up to the post-diploma level: it would be interesting to be able to establish professional training schools of excellence, capable of attracting young people from all over the world.

*"Schools in inland areas must be different from schools in the cities. In the internal areas, it is necessary to teach people to know these territories and therefore to understand their advantages and potential."* (KI10)

As highlighted several times in this discussion, certainly in the area of the "borghi", a critical issue is that of accessibility.

On the one hand, the need to launch projects aimed at encouraging barrier-free tourism is highlighted. This, however, may encounter some obstacles.

For example, in the perspective of a philological architectural recovery, if on the one hand it might be possible to make the rooms of a historic building accessible, it would be difficult to install slides on period staircases. In this sense, digital technology, in particular through VR and AR solutions, could make the cultural resources present in a "borgo" accessible.

On the other hand, it should not be underestimated that isolation is in many cases an advantage. Therefore, it is necessary to enhance accessibility without affecting the beauty of the places and the elements that make them pristine in the eyes of tourists.

In the process of enhancing the value of the "borghi", new technologies occupy a central position because, for many of those interviewed, they represent the asset that can speed up the development process and do so by generating multiplier effects.

The application of new technologies improves fruition in terms of services and activities and makes it possible to provide local residents with the same opportunities that residents of large urban centers have, eliminating discrimination.

Assuming that broadband is the basic infrastructure for using technological solutions, these solutions represent the cornerstone for supporting both technological and ecological transitions at the "borgo" level.

*"Technologies should be evaluated and considered primarily on the solutions they can provide."* (KI11)

There are numerous technological applications in this sense: from electric mobility to sustainability in which there are new-generation vehicles equipped with sensors on board that can circulate in the entire area of the village, favoring capillary movements between locations (such as in the "borgo" of Lioni), telemedicine, and health care at a distance that reassures all elderly people, whether residents or tourists, passing by the energy efficiency of buildings.

*"A fascinating world opens up, where the 'borghi' find solutions that were unthinkable before."* (KI11)

Technologies also increase the attractiveness of the area thanks to applications in tourism marketing and mainly in the use of social media.

In the final phase of this research work, the interviewees were asked to summarize, from their point of view, the threats and opportunities of the "borghi" and the possible strengths and weaknesses (see Table 1).

**Table 1.** The SWOT analysis of the "borghi": some research results.

| Threats | Opportunities |
|---|---|
| ■ Excessive individualism (KI1)<br>■ Risk of diverting funds to wrong investments (KI4)<br>■ Socio-cultural and technological distance (KI5)<br>■ Depopulation (KI5)<br>■ Indifference to the public decision-maker (KI6)<br>■ Excessive anthropization (KI7)<br>■ Touristification (KI8)<br>■ Imbalance between supply and demand (KI9)<br>■ Ignorance and selfishness to hold annuity positions (KI11) | ■ Growing attention towards "borghi" (KI1)<br>■ Revitalization processes based on territorial identity (KI2)<br>■ Rediscovery of cultural and social values (K/3)<br>■ New travel behaviors and opportunities for remote working (KI4)<br>■ Greater attention in the management of the trade-off between care and recovery of the territory and its enhancement (KI5, KI7)<br>■ Pervasiveness of the principles of sustainability in tourism (KI8)<br>■ Greater propensity to create public–private partnerships (KI8)<br>■ Desire to enjoy a small dimension, which has always been appreciated (KI9)<br>■ Attention towards technological and ecological transition (KI10)<br>■ Diffusion of Community Cooperatives (KI11) |
| Strengths | Weaknesses |
| ■ Multiplicity of characterizations of the "borghi" (KI1)<br>■ Presence of a unique cultural, material, and immaterial heritage (KI2, KI7)<br>■ Attractiveness inherent in the "borgo" concept (KI3)<br>■ Italian style and Made in as key factors in the tourism potential of our country (KI4)<br>■ Possibility of enjoying a slow visit experience (KI5)<br>■ Possibility of creating narratives (storytelling) differentiated according to the target of tourism (KI8)<br>■ Hospitable atmosphere, in which it is possible to rediscover sociality (KI10) | ■ Excessive provincialism (KI1)<br>■ Processes of standardization in proposing and communicating the "borghi" (KI2)<br>■ Inadequate promotion of inland areas (KI3)<br>■ Lack of a culture of hospitality (KI4)<br>■ Problems of seismic and health safety (KI7)<br>■ Infrastructural deficiencies (KI9)<br>■ Lack of adequate professional training (KI10) |

## 5. Conclusions and Further Research

The "borghi" are the hallmark of Italy, being spread mainly in inland areas and rural areas of the country, and are subject to the phenomenon of depopulation, demographic decline, aging of the population, and from an infrastructural point of view, strong degradation and abandonment.

The research paper contains conclusions that are developed on a twofold level. First—at the observational–interpretive level, starting from the theoretical background, the main aim was to provide a framework on the opportunities of tourism development in the are of the "borghi" and the opportunity to start a path of territorial sustainable growth.

At a second level—the regulatory level—the research could be of support and makes suggestions for some of the decisions of managers and policy makers in relation to the fashion and the conditions to be created to achieve this development.

In the first part, the research wanted to describe this scenario, while in the second part it shows how, for these "borghi", the pandemic has been a great opportunity for rebirth and repopulation derived from the practice of both proximity tourism or domestic tourism in rural and inland areas, both for remote working, which has offered a great opportunity to people to leave the big cities and work from home in these places where you can enjoy a different lifestyle from metropolitan areas thanks to the possibility of being outdoors and the general welfare derived from the relationship with people for the immediacy of human relationships.

It has been shown that the pandemic, despite the economic and social crises that hit the domestic world, has brought a great opportunity for sustainable redevelopment in these places. This goal is clearly not easily achievable if the issue is not addressed from a theoretical point of view, with clear methods, criteria, and strategies to plan a development related to sustainability. To achieve this result, as it has been shown in this study, we must start from the paradigm of territorial identity, enhancing for each "borgo" this aspect that becomes a factor of uniqueness, beauty, and authenticity. Territorial identity, in fact, is that element that unites the spatial character of the territory to its material and immaterial heritage as well as to its human dynamism. After outlining this criterion, especially for tourism purposes, each "borgo" will have its own dominant character with which to distinguish itself, but also be able to use the most abundant and usable resources of the territory and preserve the most fragile ones so that they become increasingly "smart" places; that is, attentive to the well-being of both residents and tourists.

This perspective emerged from the third part of the paper in which the empirical analysis was concentrated. In fact, thanks to the interviews of some stakeholders, important and interesting aspects have emerged that relate precisely to the future of the "borghi" and their possible tourist valorization in a sustainable vision. In fact, all the interviewees agree that it is necessary to have, especially for the tourist attractiveness, both a great technological push—in particular related to connectivity—and a new infrastructure where mobility—in particular the connections with the big urban hubs—and health and safety are the main assets on which to start the sustainable development of these places. These aspects are in fact attractors of certain tourist targets, which, as emerged from the research, can constitute not only a source of added value to the local economy, but also a major social driver that can help these "borghi" to be repopulated.

From this research, and above all, the interviews, it emerges that the valorization of the "borghi" is a very particular way to attract tourism flow because they do not attract mass tourism but a segmented tourist. For this reason, as the research explain, it is needs to rethink the tourist development of each "borgo", in which, during the new planning of tourism, the residents and the public and private stakeholders can be involved in realizing a differentiated tourist supply. During this phase of planning tourist development, each stakeholder has a role in realizing the tourist attractions, based on the abundant and local resources and cultural Heritage that the local communities know very well. The residents can participate, in fact, in the organization of all tourist products, not only seeing the landscape but in realizing the tourist experience and the events. Therefore, each "borgo"

is characterized and possesses a type of tourist demand: If we have a "borgo of water", in which there a lot of waterfalls, this borgo will be visited by families; if a "borgo" is known for medieval festivals, the tourist demand is composed by adults and families with high education levels; or if a "borgo" is known for being the birthplace of musicians, the opera and classical music festivals are welcomed and the tourists demand is composed of musicians and music lovers, etc.; if a borgo is known for its arts festivals and because a painter or famous artist was born there, the demand is composed by artists and art lovers and so on. The development of tourism will allow for the attraction of not only tourists, but a lot of young people for working in direct and indirect tourist enterprises, and thus the added value spreads throughout the region, and not only locally. However, if the tourism product is sustainable, the cycle of tourist products is not applicable and it is possible when all "borghi" are united by networking, a route of "borghi" that allow the tourists to stay for more days or repeat the visit for getting to know the other "borghi".

The research has highlighted that if the tourist development of the "borghi" is so addressed, the Italian gap between urban and rural areas can be significantly less than it is presently, but could totally disappear if infrastructures were built to interconnect territories not only digitally, but also physically through environmentally friendly mobility and sustainable infrastructure.

In conclusion, although the contribution has thoroughly analyzed the strengths, opportunities, weaknesses, and threats of the "borghi", in view of the changes taking place, further research will be necessary to verify the state of growth and the local development of the "borghi". To this end, a monitoring unit will be set up by the same authors that will verify the implementation of sustainable development for tourism in the "borghi", including interviewing other stakeholders involved in this process.

The main limitations of this study concern the qualitative method. Qualitative interviews, in fact, suffer from a high degree of subjectivity and therefore cannot be formalized. The path of qualitative research is difficult to schematize, and its results are useful to deepen a particular phenomenon but are not generalizable.

Therefore, this research is likely be only one of the first in-depth studies on this innovative and very current theme in a phase of recovery of tourism in which the experience required is of proximity, outdoor, adequately spaced, and in contact mainly with the nature and culture of the places. The research in this sense opens the way for several other insights. First of all, the opportunity to expand the number of interviewers and then go on to organize a quantitative study through a survey of both "borghi" tourists and also of tourist players that operate in the tourism offer system of these small towns.

**Author Contributions:** C.B. carried out paragraphs: Section 2 Theoretical background. R.M. carried out the Section 3 Research Method. Both the authors carried out the Abstract, the Introduction, the Results and Discussion, and the Conclusions and further research. All authors have read and agreed to the published version of the manuscript.

**Funding:** This research received no external funding.

**Institutional Review Board Statement:** Not applicable.

**Informed Consent Statement:** Informed consent was obtained from all subjects involved in the study.

**Data Availability Statement:** Data available on request due to restrictions eg privacy or ethical. The data presented in this study are available on request from the corresponding author. The data are not publicly available due to key informants gave informed consent for the use of the data and interviews only to the Authors.

**Acknowledgments:** The authors wish to acknowledge the support offered especially by all interviewees.

**Conflicts of Interest:** The authors declare no conflict of interest.

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
