# Peer review of "The Valorization of Italian “Borghi” as a Tool for the Tourism Development of Rural Areas"

_sustainability, doi:10.3390/su13126643_

Round 1

Reviewer 1 Report

Dear authors, the topic is interesting and a good premise for a study.  But, the manuscript has a number of deficiencies that need to be addressed, outlined below:

Please, follow the format specified by the journal.The proposed Sustainability Microsoft Word template file should be used.

The English needs attention. The manuscript should be revised throughout

Introduction is descriptive and does not give sufficient context. A closer connection with the existing literature needs to be established. The paper also appears to be under referenced. The main argument(s) of the paper is not clearly stated. Please give the reasons for using qualitative research method.

Lines 38-41 & 84-86: please give referencing sources

The link between community-based tourism, Italian borghi, and sustainable tourism should be clarified with literature review

The definition and description of the term Italian borghi should be more thorough. Also paradigms of villages should be given

Concepts like “sustainable tourism”, “smart workers”, “smart growth” are not clearly defined

The research approach and method are not descripted thoroughly.

In Methodology authors do not give exaples of what documents have been analyzed in order to define questions. The method section should clearly state and justify why the particular method, semistructured interviews, was chosen. Also, criteria for selecting informants should be explained and justified. An interview guide should be given.

It is not clearly stated if the findings of the qualitative research are an accurate representation of the phenomena that the paper is intended to represent

Authors should discuss results more thoroughly.  Results do not seem to have a practical implication.

The policy implications of the analysis need to be made much clearer. This will help practitioner readers to concentrate on the findings of the paper and provide them with actionable advice

Overall, despite its intention and merit, the paper is not fit for publication as it stands. I would welcome a revised version that is considerably strengthened.

Good luck!

Author Response

Reviewer #1:  Please, follow the format specified by the journal. The proposed Sustainability Microsoft Word template file should be used

Replay: Thank you for your recommendation. We have re-edited the file using the this template file.

Reviewer #1:  The English needs attention. The manuscript should be revised throughout.

Replay: Thanks for the suggestion. the English has been reviewed by a native proof reader

Reviewer #1:  Introduction is descriptive and does not give sufficient context. A closer connection with the existing literature needs to be established. The paper also appears to be under referenced. The main argument(s) of the paper is not clearly stated. Please give the reasons for using qualitative research method

Replay: Thank you for your observation. We have strengthened the introduction by emphasizing the paper aim paper, the link with the literature and the reasons for using qualitative research method.

Reviewer #1:  Lines 38-41 & 84-86: please give referencing sources. The link between community-based tourism, Italian borghi, and sustainable tourism should be clarified with literature review

Replay: We have provided references and better explained the link between community-based tourism, Italian borghi, and sustainable tourism.

Reviewer #1:  The definition and description of the term Italian borghi should be more thorough. Also paradigms of villages should be given

Replay:  Thanks for the suggestion. We have reported the concept of "Borgo" as accepted by the Encyclopedia of Tourism Management and Marketing of Prof. Buhalis, emphasizing its main features.

Reviewer #1:  Concepts like “sustainable tourism”, “smart workers”, “smart growth” are not clearly defined

Replay: Thanks for your recommendation.  We have clarified the concepts of sustainable tourism", "smart workers", "smart growth", highlighting better the link with the "borghi".

Reviewer #1:  In Methodology authors do not give exaples of what documents have been analyzed in order to define questions. The method section should clearly state and justify why the particular method, semistructured interviews, was chosen. Also, criteria for selecting informants should be explained and justified. An interview guide should be given.

Replay: Thanks a lot for your suggestion. We have integrated the whole methodological section, starting from the choice of the research method and highlighting how the key informants are selected, the interview guide and aims of this first part of the research.

Reviewer #1:  It is not clearly stated if the findings of the qualitative research are an accurate representation of the phenomena that the paper is intended to represent.

Authors should discuss results more thoroughly.  Results do not seem to have a practical implication.

The policy implications of the analysis need to be made much clearer. This will help practitioner readers to concentrate on the findings of the paper and provide them with actionable advice.

Replay:  Thank you again for this recommendation. In the Conclusions section we have included the practical implications, emphasizing how the research findings can be of support to practitioner readers.

Reviewer #1:  Overall, despite its intention and merit, the paper is not fit for publication as it stands. I would welcome a revised version that is considerably strengthened.

Replay: The revised version has been significantly strengthened.

Reviewer 2 Report

Dear Authors,

Thank you for the opportunity to review this interesting article. The topic is timely because the pandemic has caused numerous problems that local communities have to deal with, especially in regions attractive to tourists, where the life of the inhabitants was based on the provision of tourist services.

I believe that the term "borghi" should be explained in more detail. Perhaps for a reader from Italy, the meaning of this concept is obvious, but for other readers, not necessarily. However, the very well-presented empirical part has some gaps for me.

Empirical research was for: firstly, the desk phase, in which documents and information were collected from websites and concerned the cases of villages that had already taken tourism development paths that benefited local economies, second: a field phase in which 11 key informants were in-depth interviews and were semi-structured interviews. Were the questions systematized, how many questions? Have any pilot studies been carried out?

In the section Conclusions and further research, please describe further research plans, what are the research gaps, and to what extent the research will be conducted. 

Regards!

Author Response

Reviewer #2: I believe that the term "borghi" should be explained in more detail. Perhaps for a reader from Italy, the meaning of this concept is obvious, but for other readers, not necessarily. However, the very well-presented empirical part has some gaps for me.

Replay: Thanks your suggestion. We have detailed the term "Borgo" by including the definition accepted by the “Encyclopedia of Tourism Management and Marketing” of Prof. Buhalis. This aspect underlines the international relevance of the term, which has its own characterization different from the concept of village and more in line with the term "heritage town".

Reviewer #2: Empirical research was for: firstly, the desk phase, in which documents and information were collected from websites and concerned the cases of villages that had already taken tourism development paths that benefited local economies, second: a field phase in which 11 key informants were in-depth interviews and were semi-structured interviews. Were the questions systematized, how many questions? Have any pilot studies been carried out?

Replay: Thank you for this recommendation. In the research methodology section we have specified which documents, websites were consulted. We have indicated some successful cases that have undertaken tourism development paths that benefit local economies. In the description of the empirical research process, we have gone into detail about the choice of key informants, the interview guide and how the questions were developed.

Reviewer #2: In the section Conclusions and further research, please describe further research plans, what are the research gaps, and to what extent the research will be conducted.

Replay: Thanks for your observation. We have integrated the conclusion underlining the research gap that we tried to fill and the possible future developments.

Reviewer 3 Report

Comment to the authors:

Please, take into account the detailed comments below:

  1. There is room for English language improvement.
  2. line 13: there isn’t a point before of uppercase letter.
  3. line 17. There is an error in the sentence “the methodology was based on be to be interviews with open…”.
  4. line 188: “Cultural Heritage” is double repeated.
  5. The results and discussion reveal a lot of uncertainty and confusion. Probably this section must be divided into subsections in order to schematize data, strategies and so on, and clearly explained the KI, acronym not explained, outputs from the interviews.

Author Response

Dear Reviewers

Thank you for your valuable feedback and comments.

We reviewed our paper by carefully reflecting on all the received comments and suggestions.

Below, we have grouped our responses to the Reviewers’ comments. Changes are in red in the main text file. The substantial changes and content integrations addressing the Reviewers’ comments are highlighted in green.

We thank the Reviewers for your thoughtful guidance, which has helped us to improve the manuscript.

Best regards,

The Authors

Reviewer #3

  1. There is room for English language improvement.

Replay: Thanks for the suggestion. the English has been reviewed by a native proof reader.

Reviewer #3

  1. line 13: there isn’t a point before of uppercase letter.
  2. line 17. There is an error in the sentence “the methodology was based on be to be interviews with open…”.
  3. line 188: “Cultural Heritage” is double repeated.

Replay: Thanks for your observations. we have corrected these errors.

Reviewer #3

  1. The results and discussion reveal a lot of uncertainty and confusion. Probably this section must be divided into subsections in order to schematize data, strategies and so on, and clearly explained the KI, acronym not explained, outputs from the interviews.

Replay:  Thanks for your suggestions. The results and the discussion were developed according to the order of interview guideline. We did not find it necessary to divide this discussion into further subsections. However, we have explained what KIs are and have strengthened the conclusions by emphasizing the theoretical and practical implications of the work.

Reviewer 4 Report

The paper entitled “The valorization of Italian “borghi” as a tool for the tourism development of rural areas” is really interesting, even if I think a long way off being fit for publication.

The abstract must underline the originality of research. The introduction must significantly deepen the literature review in relation to sustainability, sustainable tourism in rural areas, also considering similar situations not exclusively in Italy. It is not admissible for a publication in an international Journal as Sustainability is that the entire introduction has a single reference.

In this regard, I suggest you read and quote the following papers:

  • Ivona, A., Rinella, A., Rinella, F., Epifani, F., & Nocco, S. (2021). Resilient Rural Areas and Tourism Development Paths: A Comparison of Case Studies. Sustainability 2021, 13, 3022.
  • Giordano, S. (2020). Agrarian landscapes: from marginal areas to cultural landscapes—paths to sustainable tourism in small villages—the case of Vico Del Gargano in the club of the Borghi più belli d’Italia. Quality & Quantity54(5), 1725-1744.
  • Garau, C. (2015). Perspectives on cultural and sustainable rural tourism in a smart region: The case study of Marmilla in Sardinia (Italy). Sustainability7(6), 6412-6434.

At the end of the introduction there should be an explanation by the authors on how the paper will be developed.

In Theoretical background’s section, the authors insert the concept of "smart growth policy”, without framing the concept of smartness (pag. 5). The authors do not deal in depth with the state of the art. It is dealt with in a very superficial way. The "Research Method" section is also not commented and described well. The principle of saturation is indicated without describing it in detail. In this regard I suggest you read and quote:

  • Glaser, B.G.; Strauss, A.L. The Discovery of Grounded Theory: Strategies for Qualitative Research; Aldine: New York, NY, USA, 1967.
  • Guest, G., Namey, E., & Chen, M. (2020). A simple method to assess and report thematic saturation in qualitative research. PLoS One15(5), e0232076;
  • Garau, C., & Annunziata, A. (2020). Supporting Children’s Independent Activities in Smart and Playable Public Places. Sustainability12(20), 8352. (pag 7)

Authors write “In order to pursue data saturation: (1) the interviews were structured to facilitate asking the same questions of the participants; (2) we constructed a saturation grid, in which the main issues and topics related to the conceptual framework were listed on the vertical axis and the interviews were listed on the horizontal axis [30]; (3) we relied on triangulation of data across multiple sources [31]”. In point 2, authors describe that they constructed a saturation grid. Why don't they put up a scheme, a drawing that makes it easier to understand graphically how they did it?

There is no figure in the paper. It might be useful to insert one in which all the villages/”borghi” appear.

The vast majority of citations are dated, which strongly questions the originality of the research. Extensive editing of English language and style must be required.

Author Response

Dear Reviewer

Thank you for your valuable feedback and comments.

We reviewed our paper by carefully reflecting on all the received comments and suggestions.

Below, we have grouped our responses to the Reviewer’ comments. Changes are in red in the main text file. The substantial changes and content integrations addressing the Reviewer’ comments are highlighted in green.

We thank the Reviewer for your thoughtful guidance, which has helped us to improve the manuscript.

Best regards,

The Authors

Reviewer #4

Reviewer #4

The abstract must underline the originality of research. The introduction must significantly deepen the literature review in relation to sustainability, sustainable tourism in rural areas, also considering similar situations not exclusively in Italy. It is not admissible for a publication in an international Journal as Sustainability is that the entire introduction has a single reference.

In this regard, I suggest you read and quote the following papers:

  • Ivona, A., Rinella, A., Rinella, F., Epifani, F., & Nocco, S. (2021). Resilient Rural Areas and Tourism Development Paths: A Comparison of Case Studies. Sustainability 2021, 13, 3022.
  • Giordano, S. (2020). Agrarian landscapes: from marginal areas to cultural landscapes—paths to sustainable tourism in small villages—the case of Vico Del Gargano in the club of the Borghi più belli d’Italia. Quality & Quantity, 54(5), 1725-1744.
  • Garau, C. (2015). Perspectives on cultural and sustainable rural tourism in a smart region: The case study of Marmilla in Sardinia (Italy). Sustainability, 7(6), 6412-6434

At the end of the introduction there should be an explanation by the authors on how the paper will be developed

Replay: Thanks for your observations. The introduction section has been greatly enhanced with the reported references and also integrated with the articulation of the work.

Reviewer #4

In Theoretical background’s section, the authors insert the concept of "smart growth policy”, without framing the (pag. 5). The authors do not deal in depth with the state of the art. It is dealt with in a very superficial way.

The "Research Method" section is also not commented and described well.

Replay: Thanks for your suggestion. In the Theoretical background's section we have explained better the concept of smartness and in the "Research Method" section we have explained precisely the process phases. In particular we have described how the desk research phase and the field research phase have been realized. We have explained which documents we consulted, how the key informants were chosen, how the interview guidelines were designed, how the interviews were administered and how they were processed.

Reviewer #4

Extensive editing of English language and style must be required.

Replay: Thanks for the suggestion. the English has been reviewed by a native proof reader.

Round 2

Reviewer 1 Report

The authors have satisfactorily addressed most of my comments. The revised version of the manuscript appears to be goo.

Author Response

Thank you very much for your comments.

Reviewer 3 Report

Dear authors, thank you for the revised version. There is only one mistake:

line 267: probably "is" has to be changed with "if"

Author Response

Thank you very much for your comments.

Reviewer 4 Report

the authors made all the required changes.  I only noticed that in the text, reference 3 (the new one) number 3) is indicating as Garau S. instead of Garau C.

Author Response

Thank you very much for your comments.